# Identification of Novel Cannabinoid CB2 Receptor Agonists from Botanical Compounds and Preliminary Evaluation of Their Anti-Osteoporotic Effects

**DOI:** 10.3390/molecules27030702

**Published:** 2022-01-21

**Authors:** Si-Jing Hu, Gang Cheng, Hao Zhou, Qi Zhang, Quan-Long Zhang, Yang Wang, Yi Shen, Chen-Xia Lian, Xue-Qin Ma, Qiao-Yan Zhang, Lu-Ping Qin

**Affiliations:** 1Department of Pharmacogonosy, School of Pharmaceutical Sciences, Zhejiang University of Traditional Chinese Medicine, Hangzhou 310053, China; hsj960321@163.com (S.-J.H.); gangcheng@zcmu.edu.cn (G.C.); zh995315026@163.com (H.Z.); zq199305@163.com (Q.Z.); zql20161065@zcmu.edu.cn (Q.-L.Z.); 18065148122@163.com (Y.S.); lianchenxia@163.com (C.-X.L.); 2Zhejiang Traditional Chinese Medicine &Health Industry Group CO., Ltd., Hangzhou 310016, China; wangyangsyx@163.com; 3Department of Pharmaceutical Analysis, School of Pharmacy Key Laboratory of Hui Ethnic Medicine Modernization, Ministry of Education, Ningxia Medical University, Yinchuan 750004, China; 4Department of Pharmacogonosy, School of Pharmacy, Second Military Medical University, Shanghai 200433, China

**Keywords:** cannabinoid CB2 receptor agonists, double luciferase screening system, molecular docking and dynamics, osteoblast, osteoclast

## Abstract

As cannabinoid CB2 receptors (CB2R) possess various pharmacological effects—including anti-epilepsy, analgesia, anti-inflammation, anti-fibrosis, and regulation of bone metabolism—without the psychoactive side effects induced by cannabinoid CB1R activation, they have become the focus of research and development of new target drugs in recent years. The present study was intended to (1) establish a double luciferase screening system for a CB2R modulator; (2) validate the agonistic activities of the screened compounds on CB2R by determining cAMP accumulation using HEK293 cells that are stably expressing CB2R; (3) predict the binding affinity between ligands and CB2 receptors and characterize the binding modes using molecular docking; (4) analyze the CB2 receptors–ligand complex stability, conformational behavior, and interaction using molecular dynamics; and (5) evaluate the regulatory effects of the screened compounds on bone metabolism in osteoblasts and osteoclasts. The results demonstrated that the screening system had good stability and was able to screen cannabinoid CB2R modulators from botanical compounds. Altogether, nine CB2R agonists were identified by screening from 69 botanical compounds, and these CB2R agonists exhibited remarkable inhibitory effects on cAMP accumulation and good affinity to CB2R, as evidenced by the molecular docking and molecular dynamics. Five of the nine CB2R agonists could stimulate osteoblastic bone formation and inhibit osteoclastic bone resorption. All these findings may provide useful clues for the development of novel anti-osteoporotic drugs and help elucidate the mechanism underlying the biological activities of CB2R agonists identified from the botanical materials.

## 1. Introduction

*Cannabis sativa* has been used to relieve pain and insomnia since prehistoric Eurasia Africa [1]. More than 120 cannabinoids have been isolated and identified from this plant and intensively studied for their utility in the treatment of chronic and acute pain, epilepsy and anxiety, and modulation of appetite [2,3,4]. The effects of these cannabinoids are thought to be mediated mainly by type 1 and 2 cannabinoid receptors (CB1R and CB2R), as well as other G protein-coupled receptors (GPCRs) [5]. CB1R is mainly located in the central nervous system (CNS) and is responsible for psychoactive side effects [6]. CB2R is located in immune cells and peripheral tissues, where it exerts various pharmacological activities without the psychoactive side effects induced by CB1R activation [7,8,9,10,11]. For this reason, drugs that solely target CB2R may be able to abrogate these negative side effects [12]. Therefore, CB2R agonists have attracted increasing attention and become the focus of research in recent years.

CB2R possesses a variety of physiological functions, including anti-epilepsy, analgesia, anti-inflammation, anti-fibrosis, and regulation of bone metabolism [13,14]. Bone metabolism is a bone remodeling process of osteoblastic formation and osteoclastic absorption [15]. The imbalance between bone formation and bone resorption often leads to bone loss or even osteoporotic fracture. The expression of CB2R in osteoporotic patients was lower than that in healthy bone marrow sample donors [16]. CB2R gene deficient mice were characterized by age-related trabecular loss and cortical dilation and a significant decrease in the number of trabecular and shaft osteoblast progenitor cells, while CB2 selective agonists could reduce ovariectomy-induced bone loss [17]. Both osteoblasts and osteoclasts can express CB2R. CB2R on the surface of the osteoblast membrane was found to be associated with Wnt-β-catenin pathway regulation and could increase the proliferation and differentiation of osteoblasts by enhancing the expression of runt-related transcription factor 2 (RUNX2), osteoprotegerin (OPG), and osteocalcin (OCN) [18,19]. The activation of CB2R on the membrane of osteoclasts could reduce osteoclast differentiation and bone resorption by stimulating phospholipase C, thereby regulating the intracellular calcium level and inhibiting the expression of nuclear factor κB (NF-κB) [20]. CB2R agonists are also reportedly effective drugs for treating bone loss caused by breast cancer and relieving cancer pain [21,22]. Hence, CB2 signals can be used as a molecular target for the prevention and treatment of osteoporosis [23].

Medicinal plants are a natural reservoir of secondary metabolites. The chemical entities that exert various biological activities within these plants provide versatile lead structures that are active against lots of pharmacological targets. Some botanical compounds have been developed into clinical medicines, such as artemisinin, paclitaxel, vincristine, and morphine [24]. To rapidly and efficiently identify drugs that may act as CB2R agonists, we developed a double luciferase screening system to screen CB2R agonists from 69 botanical compounds; evaluated their agonistic activity on CB2R using cAMP levels, molecular docking, and molecular dynamics simulations; and then further verified their anti-osteoporotic activities on osteoblasts and osteoclasts, hoping that the results could provide novel clues for discovering anti-osteoporotic drugs. In addition, we identified novel cannabinoid ligands to explore novel action mechanisms underlying the therapeutic effects of these compounds.

## 2. Results

### 2.1. Establishment of the Double Luciferase Screening System for Screening CB2R Modulators

After a 24-h transfection with CB2R plasmid, CB2R positive HEK293 was stably screened and identified by hygromycin B and G418. As shown in Figure 1a–c, the fluorescence intensity and the protein and mRNA expression levels of CB2R in the transfected cells were significantly increased compared with those in cell lines transferred with the empty vector only, indicating that the CB2R plasmid was stably transfected into HEK293 cells.

The performance of the constructed HEK293-CB2 cell system in screening CB2R modulators was evaluated and the results showed that the signal background value was 4.8 and the signal-to-noise ratio was 31.1, indicating that the background level had no significant effect on the result of high-throughput screening. Zhang et al. introduced a simple statistical parameter (Z’ factor) for different high-throughput screening tests [25]. Their results showed that the Z’ value of an excellent high-throughput screening model should be greater than 0.5, which can reflect the sensitivity of the screening system. It was found in our study that the signal average and standard deviation in the forskolin positive control group and DMSO blank control group were 63.3 ± 4.4% and 13.1 ± 1.6%, respectively; the signal window (SW) was 50.2%; the signal background coefficient of variation (CV) was 7.0% and 7.6%, respectively; and the Z’ factor was 0.64. These results indicate that this screening model for CB2R had good performance with high feasibility and reliability.

To verify the feasibility of the system in screening CB2R agonists, the dual luciferase activity of HEK293-CB2 cells treated with HU308, a known CB2R activator, was detected. Compared with the HEK293 cell line transfected with only an empty vector, the response of HEK293-CB2 cells to the CB2R agonist HU308 is shown in Figure 1d, demonstrating that HU308 significantly increased the relative luciferase activity. The agonistic effect of HU308 on CB2R was mitigated by AM630, an inverse agonist of CB2R (Figure 1e). The result showed that the HEK293-CB2 high-throughput screening system could be used to screen CB2 modulators from botanical compounds with high sensitivity and reliability. The luciferase activity of HEK293-CB2 cells of passages 10 (PA10), 15 (PA15), 25 (PA25), 35 (PA35), and 40 (PA40) was significantly increased after a 6-h treatment with 1 μM HU308, and there was no significant difference in the luciferase activity between different generations (Figure 1f), demonstrating that the screening system had good stability and was able to screen cannabinoid receptor modulators from botanical compounds.

### 2.2. Screening of CB2R Agonists from Botanical Compounds

To objectively evaluate the activating effect of botanical compounds on CB2R in HEK293-CB2 cells, the average double luciferase activities of HU308 and all 69 botanical compounds were measured within 6 h in three independent experiments. As shown in Figure 2a–c, 32 plant compounds showed significant activating effects on CB2R in HEK293-CB2 cells. Of the 32 compounds, only 12 compounds showed no activating effect on CB2R in HEK293-EGFP cells transfected with only EGFP (Figure 2d), indicating that the activation effect of these compounds on CB2R may be specific.

Then, the agonistic effect of these compounds on CB2R was further verified by using AM630, a CB2R inverse agonist. As shown in Figure 2e, nine plant compounds (including dihydromethysticin, desmethoxyyaxngonin, flavokawain A, echinatin, mangiferin, 11-keto-beta-boswellic acid, flavokawain C, orientin, andasperuloside) could increase the fluorescence expression in HEK293-CB2, and these effects could be abated by AM630 for 30 min, indicating that the nine compounds may be the specific potential agonists of CB2R.

### 2.3. Effects of the Screened CB2R Agonists on the Expression Level of Intracellular cAMP in HEK293-CB2 Cells

Knowing that Gαi-coupled CB2 decreased cAMP production by inhibiting the adenylate cyclase (AC) activity in cells, we used HEK293 cells that were stably expressing CB2R to perform a cAMP accumulation assay to further confirm the agonistic activity of the nine screened compounds on CB2R. HEK293–CB2 cells were pre-treated with forskolin to increase the content of intracellular cAMP and then treated with HU308 and the screened compounds, and then the changes in intracellular cAMP were measured. As shown in Figure 3, the nine botanical compounds significantly inhibited the production of forskolin-stimulated cAMP and the IC_50_ value of HU308 was 54.19 nM. As well, the IC_50_ values of flavokawain C, dihydromethysticin, desmethoxyyangonin, flavokawain A, echinatin, mangiferin, orientin, asperuloside, and11-keto-beta-boswellic acid were 5.166 µM, 35.71 nM, 29.93 nM, 0.9024 µM, 90.50 nM, 1.168 µM, 0.1106 µM, 0.2808 µM and 73.05 nM, respectively.

### 2.4. Molecular Docking Analysis of CB2R Agonists from the Botanical Compounds by Comparison with HU308

To predict the binding affinity between ligands and CB2R and characterize the different binding modes, a docking simulation was performed on the screened compounds. The results of each ligand binding with CB2R are listed in Table 1. This in silico analysis has displayed the binding affinities of the ligands and their energetic docking scores with the target. As shown in Table 1, the docking score of HU308 with CB2R was −11.604. As well, the docking scores of dihydromethysticin-1, desmethoxyyangonin, flavokawain A, echinatin, mangiferin, 11-keto-beta-boswellic acid, flavokawain C, orientin, and asperulosidewith CB2 were −8.053, −7.985, −8.508, −8.926, −13.968, −6.477, −7.769, −14.469, and −12.473, respectively, thereby demonstrating that these compounds had good affinity with CB2R.

The docking models of HU308 with CB2R agonists are shown in Figure 4a,b and Table 1. In the model, HU308 was situated in the orthosteric binding pocket, buried in the transmembrane (TM) regionsurrounded by residues from TM2, TM3, TM5, TM6, and TM7, and capped by extracellular loop2 (ECL2). The interactions between HU308 and CB2Rr were mainly hydrogen bonds, π-π interactions, and hydrophobic (Figure 4b, Table 1). The hydroxyl group on the bicyclic ring was predicted to form a hydrogen bond with the oxygen on LEU182 and the bicyclic ring had hydrophobic interactions with PHE91, PHE94, PHE106, and PHE281. The benzene ring in the middle of HU308 pointed downward and was predicted to engage in π-π interactions with PHE87 and PHE183 in a t-shape conformation. The alkyl chain of HU308 extended between TM3 and TM4 and formed hydrophobic interactions with residues from TM3, TM4, TM5, and TM6, including PHE117, TRP194, and TRP258.

As shown in Figure 4c and Table 1**,** the binding modes of representative screened compounds with CB2R were similar to that of HU308. Except for Echinatin, HU308 and other compounds (including flavokawain A, orientin, mangiferin, and asperuloside) adopted an L-shape conformation in the orthosteric binding pocket. Like HU308, flavokawain A and orientin were surrounded by many identical hydrophobic residues like PHE91, PHE94, ILE110, PHE183, and PHE281. The hydroxyl oxygen, which is on the aromatic ring of flavokawain A at the bottom of its L-shape conformation, was predicted to form a hydrogen bond with the hydroxyl hydrogen on residue SER285. Moreover, this aromatic ring also formed a π-π interactions with HIS95 in a T-shape conformation. Orientin was located at the head of the L-shaped conformation and the hydroxyl hydrogen on the aromatic ring conforms with a hydrogen bond with a hydroxyl hydrogen on residue THR114. As well, this benzene ring was also engaged in π-π interactions with PHE183 in a T-shape conformation. At the bottom of the L-shaped conformation of orientin, the hydroxyl hydrogen on the benzene ring formed a hydrogen bond with HIS95. Furthermore, the core, the naphthalene moiety of orientin, is predicted to form strong π-π interactions with PHE94. The orientation of mangiferin and asperuloside was opposite to HU308. Mangiferin and asperuloside form a hydrogen bond with residue LEU182 and SER90, and mangiferin had a π-π conjugation while asperuloside did not. Mangiferin formed three T-shaped π-π interactions with PHE183 and PHE87. As well, they also had some identical hydrophobic residues like PHE106, ILE110, TRP194, PHE183, and PHE281. Echinatin formed hydrogen bonds with residues SER285 and THR114, and also formed two π-π interactions with residue HIE95 and residue PHE183, respectively. HU308 and the screened compounds all had hydrophobic interactions with the hydrophobic residues ILE110, PHE183, and PHE281, and the screened compounds showed good affinities on CB2R, indicating that these three residues may play an important role in CB2R activation.

### 2.5. Molecular Dynamics Simulations


To further investigate the protein–ligand complex stability, conformational behavior, and interaction, molecular dynamics (MD) was performed for 10 ns by analyzing the Root Mean Square Deviation (RMSD) and Root Mean Square Fluctuation (RMSF), knowing that the lower the RMSD, the greater the stability of the protein–ligand complex. As shown in Figure 5a HU308 and Table 2, for the 5ZTY-HU308 complex, the RMSD value of the protein rose in 1 ns and then became stable, and the value fluctuated around 3 Å. This means that the protein of the complex had a stable conformation. The RMSD value of the ligand HU308 rose in 1 ns too, and it attained a maximum RMSD value of 4.2 Å at 1 ns and decreased slowly to 4 ns, and then became stable at 1.7 Å, indicating that HU308 had a stable conformation in the protein and had a stable binding to the protein. RMSF is an index of individual residue flexibility, representing how much a particular residue moves (fluctuates) during a simulation. The RMSF of the complex (Figure 5b HU308) was generally less than 2.4 Å throughout the simulation process. This also demonstrated the stability of the protein in this complex.

As shown in Table 2, the RMSD value of the protein was within 3.5 Å, and the RMSD value of the ligand compounds was within 3.8 Å, except for orientin—which was 7.0 Å—the stable time of both the protein and ligand was within 10 ns. For example, the 5ZTY-Flavokawain-A complex also performed well (Figure 5a Flavokawain A). The RMSD value of the protein and ligand both had a slight fluctuation before 3 ns and the maximum deviation of the protein and ligand was observed at 3.6 ns (3.6 Å) and 0.8 ns (3.5 Å), respectively, and then became stable; the RMSD value of the protein was around 3.0 Å and the RMSD value of the ligand was around 2.6 Å. The maximum fluctuation in RMSF was likely observed in residue ALA310 (Figure 5b Flavokawain A), which attained 3.6 Å approximately. Moreover, most of the residues were less than 2.5 Å during the simulation process. Therefore, the protein and ligand in this complex had a relatively stable conformation. Similar results were also obtained in the analysis of echinatin, mangiferin, and asperuloside (Figure 5a,b).

However, in the 5ZTY-Orientin complex (Figure 5a Orientin), there would be a mutation in the RMSD value of the protein and ligand in 0.6–0.8 ns, and the value exceeded 30 Å. Subsequently, the RMSD value of the protein and ligand dropped rapidly and tended to stabilize, and the values were around 3.0 Å and 7.0 Å, respectively. In addition, we also noted the higher RMSD value in this complex and the higher RMSD value of the ligand than the protein; as well, we noted the higher RMSF of this complex (Figure 5b Orientin) than that of HU308, with the maximum residue deviation reaching 5.6 Å and the average residues deviation being above 2.4 Å. These results suggest that the ligand may diffuse away from its initial binding site.

### 2.6. CB2R Agonists Improve the Function of Osteoblasts Derived from Bone Marrow Mesenchymal Stem Cells (BMSCs) in a CB2-Dependent Manner

Considering that CB2R was involved in the regulation of osteogenesis, we examined the effect of HU308 and the screened botanical compounds on the proliferation, ALP activity, and mineralization of osteoblasts derived from mBMSCs. As shown in Figure 6, flavokawain C, orientin, and asperuloside had no effect on the proliferation of mBMSCs cells at a concentration of 1 μM, and dihydromethysticin, desmethoxyyangonin, flavokawain A, echinatinmin, mangiferin, and 11-keto-beta-boswellic acid increased the proliferation, ALP activity, and matrix mineralization of osteoblasts derived from mBMSC. These effects could be reversed by AM630, an inverse agonist of CB2R, indicating that these botanical compounds enhanced the capacity of the osteoblasts in a CB2-dependent manner.

### 2.7. CB2R Agonists Inhibitthe Activity of Osteoclastsin a CB2-Dependent Manner

Furthermore, the effects of the screened compounds on osteoclasts derived from RAW264.7 macrophages were also determined. As shown in Figure 7a, HU308 and the screened compounds had no inhibitory effects on the viability of RAW264.7 macrophages at the concentration of 1 μM. Desmethoxyyangonin, flavokawain A, echinatin, mangiferin, and11-keto-beta-boswellic acid inhibited TRAP activity of osteoclasts derived from RAW264.7 cells induced with RANKL, as evidenced by the enzyme activity and cell staining, and these effects could be reversed by AM630 (Figure 7b,c), indicating that the screened compounds exerted inhibitory effects on the osteoclasts in a CB2-dependent manner.

The F-actin ring is a microtubule and microfilament structure in osteoclasts, playing a critical role in seal zone formation and absorption of the bone mineral matrix. The phalloidin fluorescent staining visualized the morphological characteristics of the F-actin ring. As shown in Figure 7d, treatment with HU308 and the screened compounds decreased the formation of the F-actin ring in osteoclasts, as evidenced by their thickness and completeness, and these effects could also be reversed by AM630.

## 3. Discussion

Plant cannabinoids have become attractive therapeutic compounds for the treatment of various human chronic diseases because of their high effectiveness, low toxicity, and few side effects [26]. The present study identified nine new CB2R-specific agonists and demonstrated their regulatory effects on osteoblasts and osteoclasts in a CB2-dependent manner. This may provide new leads for development of novel anti-osteoporotic drugs, and also help elucidate the action mechanism of these screened compounds.

Zhang et al. reported that a high-throughput screening method with a Z′ value greater than 0.5 was sensitive and excellent [25]. To evaluate the sensibility of the HEK293-CB2-based drug screening system, we calculated the Z′ factor for our screening system and found that the Z′ value was0.65, indicating that the sensibility of our screening method was excellent. Besides sensibility, specificity and stability are equally important to a good high-throughput screening system. The CB2R increased the luciferase activity induced by all nine compounds and was completely inhibited by AM630, indicating that CB2R activation was the sole pathway. At the same time, this system also displayed fine stability, in which HEK 293-CB2 cells from five different passages had the same response to HU308. Therefore, we successfully established a good high-throughput screening system for detecting CB2 agonists from botanical compounds.

HEK293-CB2 cells were utilized to screen CB2R agonists from botanical compounds, and 69 compounds showed significant agonistic effects on CB2R. In order to exclude the possibility of false positive results, HEK293-EGFP cells were only transfected with EGFP, and AM630 was used as aCB2R inverse agonist to further confirm the agonistic effect of primary screened compounds on CB2R. Finally, nine botanical compounds were identified as CB2R agonists. Furthermore, the activation of CB2R led to the inhibition of AC via Gαi subunits, causing a decrease in 3′,5′-cyclic adenosine monophosphate (cAMP) [27,28]. We then examined the inhibitory effects of the screened compounds on levels of cAMP and further confirmed the agonistic effects of the screened compounds on CB2R. These compounds had diverse chemical structures, including flavonoid (Mangiferin, Orientin), chalcone (echinatin, flavokawain A, flavokawain C), triterpenoid (11-Keto-beta-boswellic acid), iridoid glycosides (asperuloside), and lactone (desmethoxyyangonin, dihydromethysticin).

Molecular docking is a theoretical simulation method that mainly studies the binding mode and affinity between molecules (such as ligands and receptors). Through the prediction of the affinity, as represented by the docking score, we can compare the affinity of each ligand and protein quickly so a large number of compounds can be screened successfully [29]. In this study, we investigated the binding mode and affinity between molecules and CB2R by comparison with those of HU308 and found that the screened compounds had good affinity with CB2R. However, the activities on CB2R, based on the docking scores, are not consistent with the results characterized by the double luciferase activities and the level of alteration of cAMP in HEK293-CB2 cells. This may be further investigated through experimental study. In addition, the top five compounds, including flavokawain A, echinatin, mangiferin, orientin, and asperuloside, had similar binding modes with HU308, and they all had hydrophobic interactions with the hydrophobic residues ILE110, PHE183, and PHE281. These results suggest that the three residues may play an important role in CB2R activation.

The molecular dynamics simulation, which shows the microscopic motion change process of the complex system at the atomic level, like organic solids, protein–ligand complexes, and ribosomes, can intuitively characterize the binding process, mechanism, and law of natural compounds with CB2R [30]. RMSD and RMSF values were used to characterize the degree of conformational change of the protein and ligand, the binding stability, and the degree of conformational change of the amino acid residues on the protein chain. Our results demonstrated that flavokawain, echinatin, mangiferin, and asperuloside had similar results with HU308. The RMSD value and RMSF value of the CB2 protein and echinatin/asperuloside in the complex were both less than 3.0 Å, and the curves of the protein and the ligand RMSD values during the simulation were converged and stable, indicating that the natural ligand induced the least conformational change and stability of the 5ZTY-ligand interactions. However, the simulation results of the orientin indicated that the ligand may be diffused away from its initial binding site, so the activating effects of orientin on CB2R need further investigation.

CB2R is expressed on the surface of osteoblasts and osteoclasts, and CB2R activation in osteoblasts and osteoclasts stimulates bone formation and inhibits bone resorption [17,31]. We evaluated the regulatory effect of the screened botanical compounds on osteoblasts and osteoclasts and found that dihydromethysticin, desmethoxyyangonin, flavokawain A, echinatin, mangiferin, and 11-keto-beta-boswellic acid could improve the bone formation of osteoblasts, as evidenced by the proliferation, ALP activities, and mineralization of the bone matrix, and desmethoxyyangonin, flavokawain A, echinatin, mangiferin, and11-keto-beta-boswellic acid suppressed the bone resorption of the osteoclasts, as revealed by the TRAP activities and morphologic characteristics of the F-actin ring of osteoclasts. The effects of the above botanical compounds on osteoblasts and osteoclastsmay be mitigated by AM630, which further supports the agonistic effect of these botanical compounds on CB2R. Some investigations have reported the regulatory effects of 11-keto-beta-boswellic acid, mangiferin, and orientin, and our results further revealed that these compounds may modulate bone metabolism by targeting CB2R [32,33,34,35,36,37,38,39]. To the best of our knowledge, this is the first study reporting the effects of dihydromethysticin, desmethoxyyangonin, flavokawain A, echinatin, and11-keto-beta-boswellic acid on osteoblasts and osteoclasts. HU308 as a selective CB2R agonist has been shown to decrease bone loss in an osteoporotic animal model, such as ovariectomized rats and spinal cord injury rats [40]. In addition, it also exerted a regulatory effect on the osteoblasts and osteoclasts in a CB2R-dependent manner [17,41]. Furthermore, some CB2R agonists from natural products, such as (E)-β-caryophyllene, magnolol, 4-O-methylhonokiol, celastrol, resveratrol, betulinic acid, anthocyanins, and biochanin A have been shown to have potential effects on bone metabolism. These findings, together with our results, provide potential leadsfor anti-osteoporotic drugs targeting CB2R.

In addition, the agonistic effects of these botanical compounds on CB2R may helpexplain and understand their biological activities. Desmethoxyyangonin, dihydromethysticin, flavokawain A, and flavokawain C, as the major chemical constituents of *Piper methysticum* Forst., have exhibited significant anti-inflammatory, anti-cancer, and neuro-protective effects [42,43,44,45,46,47,48,49,50,51]. 11-keto-β-boswellic acid is a triterpenoid found in Boswellia serrata and has potential therapeutic efficacy against several chronic inflammatory diseases and neurodegenerative diseases [52,53,54,55,56,57,58,59,60]. Asperuloside, which is classified under the category of iridoid glycosides, has potential for the treatment of obesity, inflammatory diseases, cancer, bacterial infection, and renal fibrosis [61,62,63,64,65,66]. Echinatin, as a bioactive flavonoid of Glycyrrhiza plants, displays antioxidant and anti-inflammatory activities [36,67,68]. Mangiferin is a xanthone present in significant levels in higher plants and possesses a variety of pharmacological actions, including anticancer, antidiabetic, antioxidative, immunomodulatory, hepatoprotective, analgesic, and neuroprotective effects [69,70]. Orientin is a water-soluble flavonoid C-glycoside with antioxidant, antiaging, antiviral, antibacterial, anti-inflammation, vasodilatation and cardioprotective, antiadipogenesis, antinociceptive, radiation-protective, neuroprotective, and antidepressant-like effects [71,72,73,74]. CB2R was also reported to be involved in the regulation of inflammation, anti-cancer, anti-fibrotic, and nerve system disease [10,13]. Therefore, our findings may provide new clues for developing these botanical compounds into cannabinoid CB2R agonists.

## 4. Materials and Methods

### 4.1. Reagents

Reagents used in this study were human embryonic kidney cells (Stem Cell Bank of the Chinese Academy of Sciences, Shanghai, China), pIRES2-EGFP-CB2, 30%Acrylamide, Ammonium persulphate (APS), Thermo Scientific Pierce (TEMED), NOT-Fat Powdered Milk, Sodium dodecyl sulfate (SDS), NaCl, Tween-20, and Tris (Hydroxymethyl) Aminomethane (Shenggong Biological Engineering Co., Shanghai, China).As well, we used pIRES2-EGFP, pGL4.29 [luc2P/CRE/Hygro], pRL-TK9 luciferase reporter Plasmid (Bei Na Biology co., Beijing, China), fetal bovine serum (FBS), Dulbecco’s modification of Eagle’s medium (DMEM), high-glucose medium, and penicillin-streptomycin double-antibody (Biological Inc, Beijing, China). Other reagents used were lipofectamine 3000 Transfection reagent, Pierce™ ECL Western Blotting Substrate, Thermo Fisher/Pierce BCA Protein Assay Kit and molecular weight standard of pre-dyed protein (Thermo Fisher Scientific Inc., Waltham, MA, USA),PrimeScript™ RT reagent Kit with gDNA Eraser and TB Green^®^ Fast qPCR Mix (Takara Bio, Beijing, China), PMSF protease inhibitor, RIPA Protein lysate, and 1.5 M Tris-HCl pH 8.8, 1 M Tris-HCl pH 6.8 and 10% SDS double luciferase reporter gene detection kit (Biyuntian Biotechnology Co., Shanghai, China). Also used was a SDS-PAGE protein loading buffer (5×) (Ford Biotechnology Co., Hangzhou, China), Immobilon-P Transfer Membrane/0.45 μm (Millipore, Billerica, MA, USA), goat anti-rabbit IgG-HRP antibody (Cell Signaling Technology Biotechnology Co., Shanghai, China), HU308 and forskolin (GLPBIO Co., Montclair, CA, USA), CCK-8 assay kit (Meilun Biopharmaceutical Co., Dalian, China), human cyclic adenosine monophosphate (cAMP) assay kit (Enzyme Immunity Biology Co., Wuhan, China), Rochelle salt (Yuanye Biotechnology Co., Shanghai, China), 4% paraformaldehyde (Seville Biotechnology Co., Wuhan, China), alkaline phosphatase (ALP) kit (Jian Cheng Technology Co., Nanjing, China), and a tartrate resistant acid phosphatase (TRAP) staining kit (Sigma-Aldrich Co., St. Louis, MO, USA). The botanical compounds used in this study were prepared in our laboratory or supplied by Professor Ma Xueqin from Ningxia Medical University (Ningxia, China), and their purity was more than 98%, as analyzed by the HPLC method.

### 4.2. Cell Culture

Human fetal kidney-derived cells (HEK293 cells) were cultured in DMEM containing 10% FBS and 1% penicillin-streptomycin double-antibody at 37 °C with 5% CO_2_.

BMSCs were isolated from the femur of 1-day-old Wistar rats and cultured in a dish containing α-MEM supplemented with 10% FBS, 100 U·mL^−1^ penicillin, 100 mg·mL^−1^ streptomycin, and 1% L-Glutaminea in a humidified atmosphere of 5% CO_2_ for 24 h. The medium was changed every three days. The subculture cells (passage 2–4) were used for subsequent experiments. BMSCs were identified in terms of the morphological character and properties of differentiation into osteoblasts. This experiment was approved by the Bioethics Committee of Zhejiang University of Traditional Chinese Medicine.

The murine macrophage RAW264.7 cell line, which has the potential to differentiate into osteoclasts under the induction of RANKL, was obtained from the Cell Bank of the Chinese Academy of Sciences (Shanghai, China). The cells were cultured in DMEM containing 10% (*v*/*v*), 1% (*v*/*v*) penicillin, and streptomycin (Beyotime, Shanghai, China) at 37 °C in humidified air with 5% CO_2_, and then subcultured every two days. For induction of the osteoclasts, RAW264.7 cells were incubated with 25 ng·mL^−1^ RANKL for 72 h. Osteoclasts were identified by TRAP staining.

### 4.3. Establishment of CB2R Double Luciferase Screening System

#### 4.3.1. Construction of CB2-Specific Reporter Plasmid

The primer pairs of the CB2R marker genes were designed by using Primer Express software (Application Biosystems) and using a GAPDH gene as an internal control. Thesingle-stranded CB2R oligomers of the EcoR I bonding end and BamH I binding end were synthesized and annealed to form double-stranded molecules. The double-stranded oligomers were digested by EcoR I and BamH I, ligated with T4DNA ligase, and cloned upstream of the luciferase coding region on the pIRES2-EGFP vector.

#### 4.3.2. Establishment of HEK293-CB2R Cells

To generate the HEK293-CB2 cells, 5 × 10^5^ HEK293 cells were cultured in a 60 mm culture dish for 12 h, washed with PBS, and then transfected with Lipofectamine 3000 reagent containing a test reporter plasmid, control reporter plasmid encoding Renilla luciferase, and effector plasmid. The ratio of the transfection plasmid pIRES2-EGFP-CB2, pGL4.29 [luc2P/CRE/Hygro], and pRL-TK was 2 μg, 10 μg, and 1μg, respectively. After a 24-htransfection, cells were selected and cultured in the medium containing G418 (800 μg·mL^−1^) and hygromycin (400 μg·mL^−1^) for two weeks. The surviving cells were picked and expanded as HEK293-CB2R cells.

#### 4.3.3. Fluorescence Quantitative Reverse Transcriptase-Polymerase Chain Reaction (FQ-RT-PCR)

The RNA of HEK293-EGFP and HEK293-CB2 cells were isolated with TRIZOL reagent, dissolved in DEPC water, and stored at −80 °C for use. The primer pairs of the CB2R marker gene were designed using Primer Express design software (Application Biosystems) and used a GAPDH gene as the internal control. The primer sequences are as follows: for CB2R, F 5′-CCGAATGAGGCTGGATGT-3′ and R 3′-ATGGAGCAGAAAGCAAAGG-5′; for GAPDH, F 5′-TATCGGACGCCTGGTTAC-3′ and R 3′-TCCTGGAAGATGGTGATGG-5′. An FQ-RT-PCR was performed for 40 cycles on a real-time PCR instrument, including a 95 °C denaturation, 30 s annealing at 59 °C, and a 30 s extension at 72 °C. All amplifications and detection were carried out in MicroAmp optical 384-well plates with optical binders. The relative expression level of CB2R was calculated by 2-CT. The relative expression (%) = 2″Δ CT × 100%, where CT represents the threshold period and Δ CT = CT (CB2) − CT (GAPDH).

#### 4.3.4. Western Blot Analysis

The total protein of HEK293-EGFP and HEK293-CB2 was extracted with protein extraction kits. The protein concentration was analyzed using the BCA protein determination kits. The protein complexes were separated by SDS-PAGE and transferred to the PVDF membrane. After sealing with 5% skimmed milk (Mamp V), the membrane was incubated overnight with anti-CB2R (1:500) and GAPDH (1:1000), respectively, at 4 °C. The PVDF membrane was then incubated with an anti-rabbit secondary antibody (1:1000) at 37 °C for 1 h. Finally, the protein imprinting was observed by a chemiluminescence detector, and the intensity was analyzed with a MONAD QuickChemi 5100 chemiluminescence imaging system.

#### 4.3.5. Assay Optimization

Several aspects of the high-throughput screening system were optimized to ensure efficient and accurate application of the system for high-throughput data collection. To conduct high-throughput assays in 96-well plates, the number of cells seeded per well, incubation time, drug and AM630 concentrations, and the DMSO content were optimized. The dual luciferase assay was performed as described previously. The result showed that the signal-to-background (S/B) ratio was the highest when 5 × 10^5^ HEK293-CB2 cells were seeded per well for a 6-h incubation period (Figure 1a,b) When the drugs dissolved in a medium supplemented with Dimethyl sulfoxide (DMSO) (≤1‰)—the optimal concentration of the natural compound—plant extract, and Forskolin, which were 10^−6^ M, 100 μg·mL^−1^, and 1 μM, respectively (Figure 1d–f). Subsequently, the quantitative evaluation indexes of the high-throughput screening system, including the signal-to-background ratio, S/B = M_signal_/M_background_; signal window, SW = M_signal_ − M_background_; signal-to-noise ratio, S/N = (M_signal_ − M_background_)/SD_background_; signal background coefficient of variation, CV(%) = 100 × SD_signal_/M_signal_; and Z’ factor were determined to ensure its efficient and accurate application of the system in high-throughput data collection. Here, Z′ = 1 − 3(SD_signal_ + SD_background_)/|M_signal_ − M_background_| [25], where M represents an average of the relative luminescence value and SD is the standard deviation of M.

#### 4.3.6. Stability Evaluation of HEK293–CB2 Cells

The HEK293-CB2 cells from 10–40 passages were separately inoculated into 96-well plates at 5 × 10^3^ cells per well. After 12 h, the cells were treated with 1 μM HU308 for another 6 h and the double luciferase activity was determined. The results were expressed as relative luminescence values.

### 4.4. Screening of Botanical Compounds

The HEK293-CB2 cells were inoculated on a 96-well culture plate with a density of 5 × 10^3^ cells per well. After the cells were adhered and starved for 24 h, the HEK293-CB2 cells were treated with 1 μM HU308 and 69 plant monomer compounds for 6 h. The relative luminescence value was used to express the expression of CB2R.

The compounds that activated the expression of CB2R were further examined on the HEK293-EGFP cells to exclude the false positive compounds. Briefly, the HEK293-EGFP cells were inoculated on a 96-well culture plate with a density of 5 × 10^3^ cells per well. After cell confluence and 24-h starvation, the HEK293-EGFP cells were treated with 1 μM HU308 and 32 plant monomer compounds for 6 h. The expression of CB2R is expressed by the relative luminescence value.

Furthermore, the activation of the compounds on CB2R was also verified by using the CB2R inverse agonist AM630. The HEK293-CB2 cells were inoculated on a 6-well culture plate at a density of 5 × 10^4^ cells per well. After adhering to the wall, the cells were starved for 24 h, pretreated with 1 μM AM630 for 30 min, and treated with 1μM HU308 and 12 natural compounds (dihydromethysticin, desmethoxyyangonin, flavokawain B, flavokawain A, neohesperidin, echinatin, baicalin, mangiferin, 11-keto-beta-boswellic acid, flavokawain C, orientin, and asperuloside) for 6 h. The expression of the CB2R gene is expressed by the relative luminescence value.

### 4.5. Determination of Levels of Cyclic Adenosine Monophosphate (cAMP) in Cells

The HEK293-CB2 cells were cultured on a 6-well plate at a density of 5 × 10^4^ cells per well until they adhered to the wall and then they were starved for 24 h, pretreated with 1 μM forskolin for 30 min, and then treated with 1 μM HU308 and the screened botanical compounds (dihydromethysticin, desmethoxyyangonin, flavokawain A, echinatin, mangiferin, 11-keto-beta-boswellic acid, flavokawain C, orientin, and asperuloside) for 6 h. After treatment, the cell supernatant was diluted with PBS (pH 7.2–7.4). The cells were lysed by repeated freeze–thaw cycles and being centrifuged for 20 min at 2000–3000 r.p.m. The concentration of cAMP in the cell lysate was determined according to the instructions of the cAMP kit.

### 4.6. Molecular Docking

The X-ray crystal structure of the CB2 protein complex was obtained from the Protein Data Bank (PDB code: 5ZTY) [75]. The docking program Autodock Vina (Version 1.1.2) (Scripps Research, San Diego, CA, USA) was used to predict the ligand-binding pose for the CB2 protein [76]. The pre-processing of the protein was carried out by AutoDockTools(AutoDockTools, version 1.4.5) (Scripps Research, San Diego, CA, USA)to remove the solvent and original ligands in the complex. It was then centered on the original ligand and a docking box was generated within 15 Å. Pre-processed ligands derived from ChemBio 3D had a minimized energy with an added hydrogen and were assigned a Gasteiger charge. The docking experiments were carried out in a semi-flexible way with only the ligands being flexible. The obtained docked pose was analyzed with AutoDockTools and PyMOL (https://pymol.org/2/, accessed on 18 January 2022). The final result was depicted by PyMOL.

### 4.7. Molecular Dynamics Simulation

The molecular dynamics simulations (MD) were carried out using the Amber 12, which mainly consists of four programs: Leap, Antechamber, Sander, and Ptraj [77]. First, Tleap in program Leap was used to generate the topology and coordinate files of the receptor protein 5ZTY and was followed by the use of different force fields to calculate the parameters of the corresponding biomolecules, where force field ff14SB was used for 5ZTY, force field lipid 14 for the lipid bilayer, and force field GAFF for all six ligands (agonist HU308 and predicted) [78,79,80]. The TIP3P water model was used to surround and fill the complex (with the water boundary box set to a box size of X = 10 Å, Y = 10 Å, and Z = 10 Å) [81]. The counter ions were added to neutralize the complex system. Asa newly built system is often accompanied by local unreasonable phenomena, such as the distance between adjacent atoms, the Sander program was used to optimize the energy. The isothermal–isobaric (NPT) ensemble was used, and the temperature was maintained at a constant 300 K. The molecular dynamics simulation run time was set at 10 ns with a recording interval of 50 ps and an energy of 1.2 ps. The results were analyzed according to the Root Mean Square Deviation (RMSD) and Root Mean Square Fluctuation (RMSF).

### 4.8. Determination of Proliferation, ALP Activities, and Bone Mineralized Nodules of Osteoblasts

The BMSCs cells were inoculated in a 96-well plate at a density of 1 × 10^4^ cells per well and cultured at 37 °C for 24 h. The cells were treated with 1 μM HU308 and the nine screened natural CB2 receptor agonists (dihydromethysticin, desmethoxyyangonin, flavokawain A, echinatin, mangiferin, 11-keto-beta-boswellic acid, flavokawain C, orientin, and asperuloside) for 48 h, and 10 μL CCK-8 solution was added to the culture well and incubated for 30 min. The optical density (OD) value was determined at 450 nm. The BMSCs cells were treated as described above and cultured for seven days to measure the ALP activity according to the kit instructions or cultured for five days to stain ALP in the osteoblasts according to the following procedures. Briefly, cells were washed with pre-cooled PBS twice and stained with the alkaline phosphatase staining solution for 30 min at 37 °C. After discarding the dye solution, cells were rinsed with distilled water three times and observed and photographed under a light microscope.

Cells were treated as described above for 14 days to observe the bone mineralized nodule formation using alizarin red staining solution. Briefly, the cells were washed with pre-cooled PBS twice, fixed with 3.7% formaldehyde for 10 min, and stained with ARS dye (pH = 4.2) containing 40 mM alizarin redstain for 10 min. The morphology and number of bone mineralized nodules of osteoblasts were observed under a light microscope.

### 4.9. Determination of Viability and TRAP Activityin Osteoclasts

RAW264.7 cells were inoculated in a 96-well plate at a density of 1 × 10^4^ cells per well. After a 24-h culture at 37 °C, cells were treated with the test compounds for 48 h and cell viability was determined with the CCK-8 kit according to the instructions. RAW264.7 cells were inoculated on a 96-well culture plate at a density of 1 × 10^4^ cells per well, induced with 25 nM of nuclear factor-κB receptor activator ligand (RANKL) to differentiate into osteoclasts for 72 h. After three-day treatment of cells with 1 μM HU308 and the screened compounds, the TRAP activity of the osteoclasts was measured. Briefly, the cells were washed with sodium thiobarbital twice, lysed with 0.1% Triton X-100 (10 μL per well) for 10 min, and then incubated with 100 μL of TRAP reaction solution (0.02 g disodium nitrophenyl phosphate and 0.1 g potassium sodium tartrate dissolved in 10 mL deionized water with pH adjusted to 3.5 by addition of 1 mol·L^−1^ HCl) for 30 min at 37 °C. The reaction was terminated with 1 mol·L^−1^ of NaOH solution (100 μL per well) and the absorbance was determined at a wavelength of 405 nm. The TRAP activity was expressed by the amount (mole) of p-nitrophenol produced by 100 osteoclasts. For staining, the osteoclasts were treated as described above, stained according to the instruction of the TRAP staining kit, and observed and photographed under a light microscope.

### 4.10. F-Actin Ring Formation Assay

RAW264.7 cells were incubated in a confocal culture dish (diameter, 35 mm) at adensity of 5 × 10^4^ cells·dish^−1^ for 24 h, treated with 25 ng·mL^−1^ RANKL and the screened compounds for five days, fixed with 4% paraformaldehyde for 30 min, stained with 5 μg·mL^−1^ rhodamine-conjugated phalloidin (Sigma-Aldrich Chemical Co.) at 37 °C for 40 min, washed with PBS again, and stained again with 4′,6-diamidino-2-phenylindole (DAPI, Beyotime, Shanghai, China) for 10 min at room temperature (Li et al., 2018). The F-actin ring of the osteoclasts was observed under a fluorescent microscope (Axiovert 200, Carl Zeiss, Oberkochen, Germany). The fluorescence images of the cells were processed using Zeiss ZEN software.

### 4.11. Statistical Analysis

The data were analyzed by the prism 9.0 software package, and the multiple comparison data were analyzed by a single-factor analysis of variance and Dunnettt-test. The value of *p* < 0.05 was considered statistically significant and all results are expressed as the mean ± standard deviation (SD).

## 5. Conclusions

This study identified nine novel CB2R agonists from botanical compounds, which specifically increasedthe expression of CB2R, and inhibited cAMP accumulation stimulated by Forskolin in HEK293-CB2 cells. In addition, these CB2R agonists exhibitedgood affinity with protein, as predicted by the molecular docking and molecular dynamics. Of these compounds, desmethoxyyangonin, flavokawain A, echinatin, mangiferin, and 11-keto-beta-boswellic acid showed significant effects on osteoblastic bone formation and osteoclastic bone resorption (Figure 8). Collectively, our findings suggest that these CB2R agonists may be potential therapeutic agents for osteoporosis, though further studies are required to further verify conclusions.

## Figures and Tables

**Figure 1 molecules-27-00702-f001:**
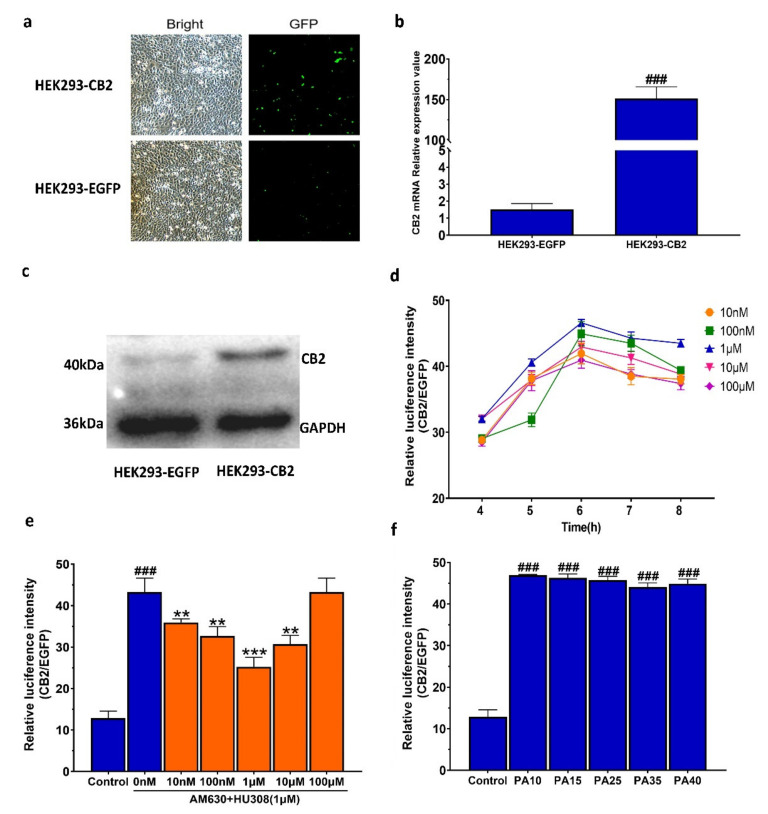
Establishment of the double luciferase screening system for screening CB2R modulators. (**a**) HEK293 with or without transfected CB2R (×100, Scale bar: 50 μm). (**b**) CB2 mRNA expression in HEK293 with or without transfected CB2R. The data shown are from six independent experiments and expressed as mean ± SD (*n* = 6). ### *p* < 0.001 compared with untransferred cell lines. (**c**) CB2 protein expression in HEK293 with or without transfected CB2R, using GAPDH as loading control. (**d**) The agonistic effect of HU308 on CB2R in HEK293-CB2 cells. HEK293-CB2 cells were treated with HU308 at concentrations of 10 nM~100 µM for 4–8 h and relative luminescence values were used to indicate the agonistic effect of HU308 on CB2R. (**e**) The reversal effect of AM630 on the agonistic activity of HU308. (**f**) The agonistic effect of HU308 on HEK293-CB2 cells with different passages. The data shown are from six independent experiments and expressed as mean ± SD (*n* = 6). ### *p* < 0.001 compared with the control. ** *p* < 0.01 and *** *p* < 0.001 compared with HU308 treatment group. GFP, green fluorescent protein; CB2, type 2 cannabinoid; EGFP, enhanced green fluorescent protein; HEK293-EGFP, the HEK293 cells only transferred into the empty vector(pIRES2-EGFP); HEK293-CB2, the HEK293 cells transferred into pIRES2-EGFP and CB2 gene; GAPDH, glyceraldehyde-3-phosphate dehydrogenase; PA, passage.

**Figure 2 molecules-27-00702-f002:**
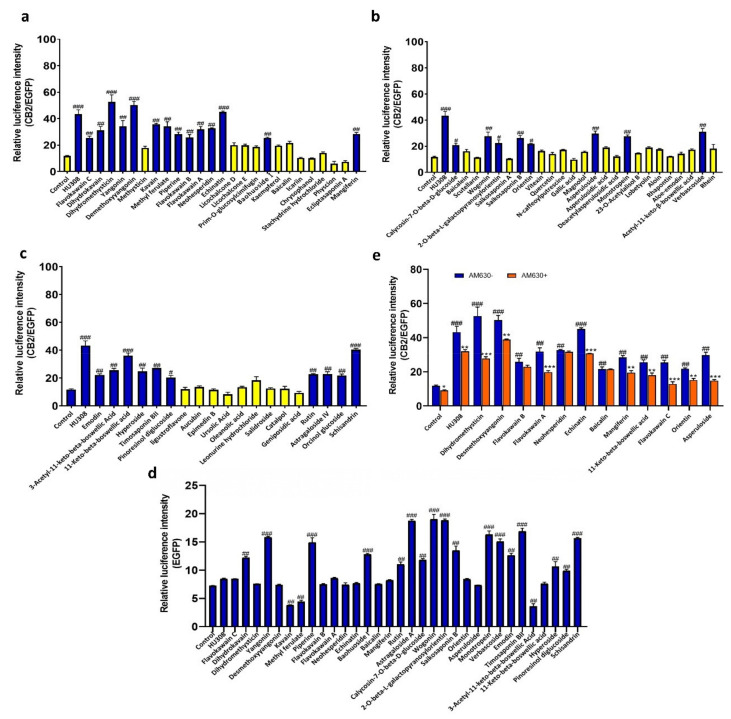
The agonistic effect of botanical compounds on CB2R based on double luciferase reporter assay. (**a**–**c**) Effects of botanical compounds on CB2R activation in HEK293-CB2 cells. HEK293-CB2 cells were treated with botanical compounds for 6 h and relative luminescence values were used to indicate the agonistic effect of the botanical compounds on CB2R. (**d**) Effects of the botanical compounds on CB2R activation in HEK293-EGFP cells. (**e**) The reversal effect of AM630 on the agonistic activity of CB2R in HU308 and screened compounds in HEK293-CB2 cells. The data shown are from three independent experiments and expressed as mean ± SD (*n* = 3). # *p* < 0.05, ## *p* < 0.01 and ### *p* < 0.001 compared with the control. * *p* < 0.05, ** *p* < 0.01 and *** *p* < 0.001 compared with the corresponding compounds treatment group. EGFP, enhance green fluorescent protein; CB2, type 2 cannabinoid.

**Figure 3 molecules-27-00702-f003:**
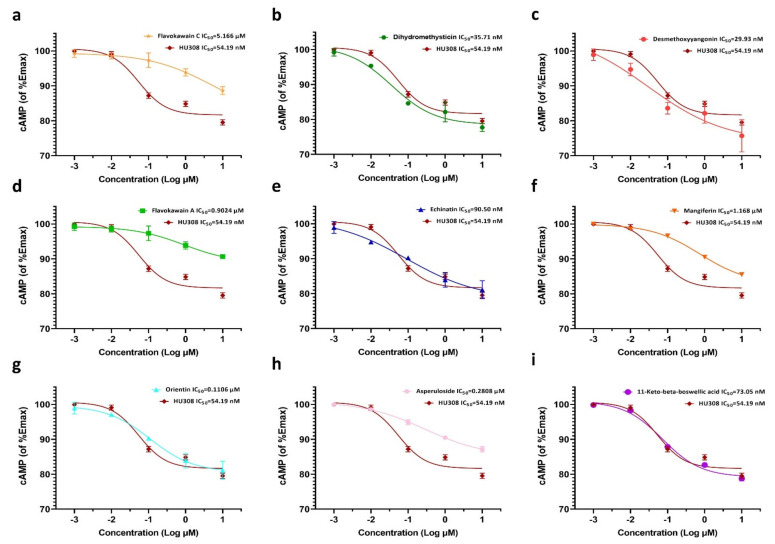
Effects of the screened compounds on intracellular cAMP accumulation in HEK293-CB2 cells. HEK293-CB2 cells were pretreated with forskolin for 30 min and then treated with the screened compounds for 6 h. cAMP level in HEK293-CB2 cells was determined with the cAMP assay kit according to the manufacturer’s instructions. (**a**) Flavokawain C. (**b**) Dihydromethysticin. (**c**) Desmethoxyyangonin. (**d**) Flavokawain A. (**e**) Echinatin. (**f**) Mangiferin. (**g**) orientin. (**h**) Asperuloside. (**i**) 11-keto-beta-boswellic acid. The data shown are from three independent experiments and expressed as the mean ± SD (*n* = 3). cAMP, cyclic adenosine monophosphate.

**Figure 4 molecules-27-00702-f004:**
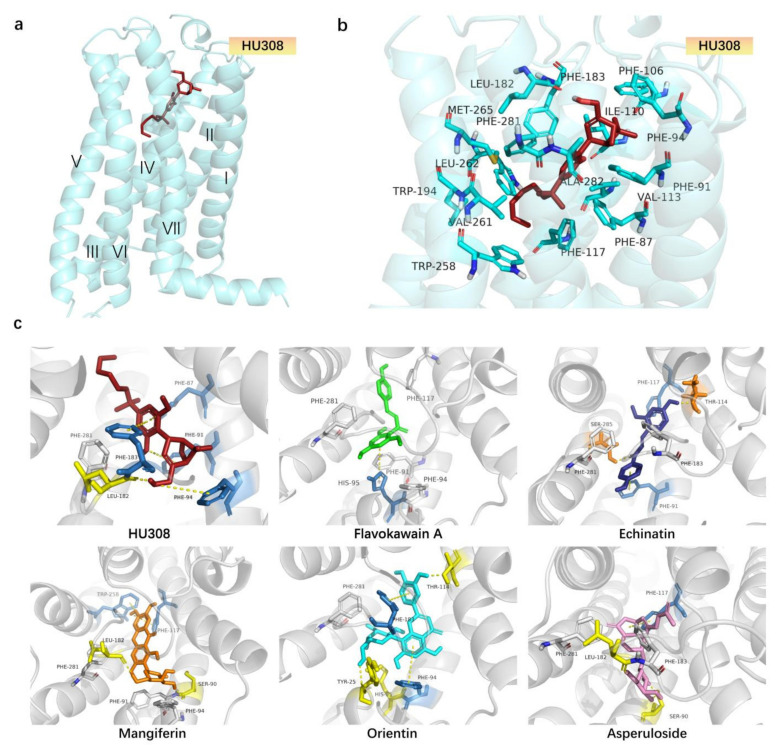
Docking poses of HU308 and screened CB2R agonist in CB2R protein. (**a**) Overall structure of the 5ZTY-HU308 complex. Protein 5ZTY is shown in cyan, with the ligand HU308 in firebrick. (**b**) Binding pose of HU308 with CB2 protein 5ZTY. Key residues, cyan sticks; involved in HU308, firebrick sticks; binding in CB2-Gi complex structure, cyan cartoon. (**c**) The binding poses of the screened compounds in CB2 protein 5ZTY, among which, HU308, firebrick sticks; flavokawain A, green sticks; echinatin, deepblue sticks; mangiferin, orange sticks; orientin, cyan sticks; asperuloside, pink sticks; hydrogen bonded residues, yellow sticks; residues with π-π interactions, skyblue sticks; conservative residues, gray sticks.

**Figure 5 molecules-27-00702-f005:**
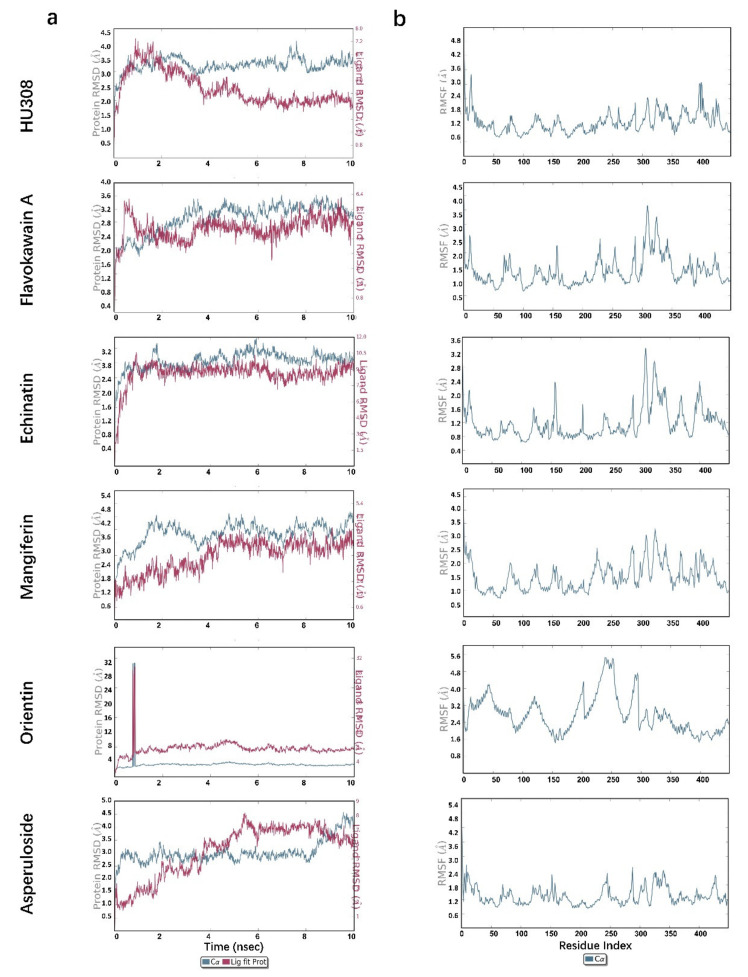
Molecular dynamics analysis of HU308 and representative screened compounds with CB2 protein. (**a**) RMSD analysis and (**b**) RMSF analysis.

**Figure 6 molecules-27-00702-f006:**
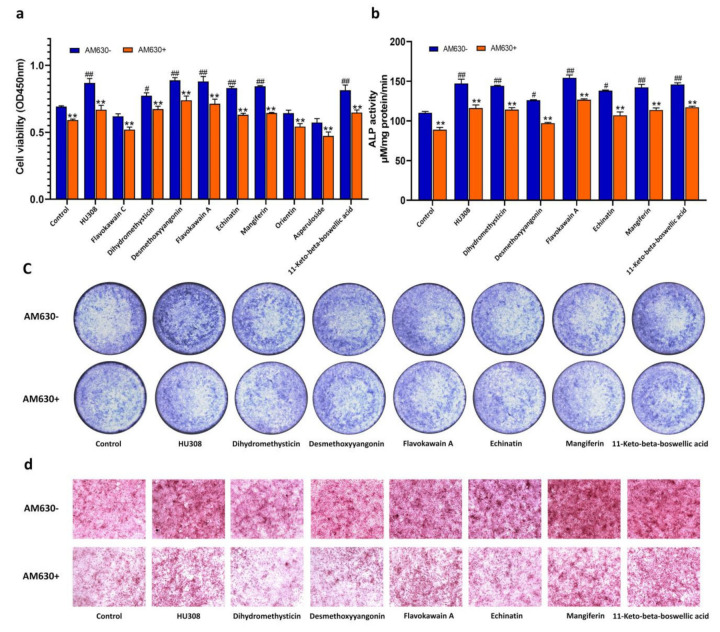
Effects of the screened compounds on the function of osteoblasts derived from bone marrow mesenchymal stem cells (BMSCs). (**a**) Proliferation of BMSCs. After 48-h treatment of BMSCs with HU308 and screened compounds, cell viability was detected by CCK8. (**b**) ALP activity. BMSCs were treated for seven days in the presence or absence of 1 μM AM630, ALP activities were measured by p-nitrophenyl sodium phosphate method. (**c**) ALP staining. The BMSCs cells were treated for five days in the presence or absence of 1 μM AM630, ALP activity was stained with BCIP/NBT assay kit. (**d**) Bone mineralization nodule staining. BMSCs were treated with screened compounds for 14 days in the presence or absence of 1 μM AM630, the bone mineralization nodules were detected by alizarin red staining. The data shown are from three independent experiments and expressed as mean ± SD (*n* = 3). # *p* < 0.05 and ## *p* < 0.01 compared with the control. ** *p* < 0.01 compared with the corresponding compounds treatment group.

**Figure 7 molecules-27-00702-f007:**
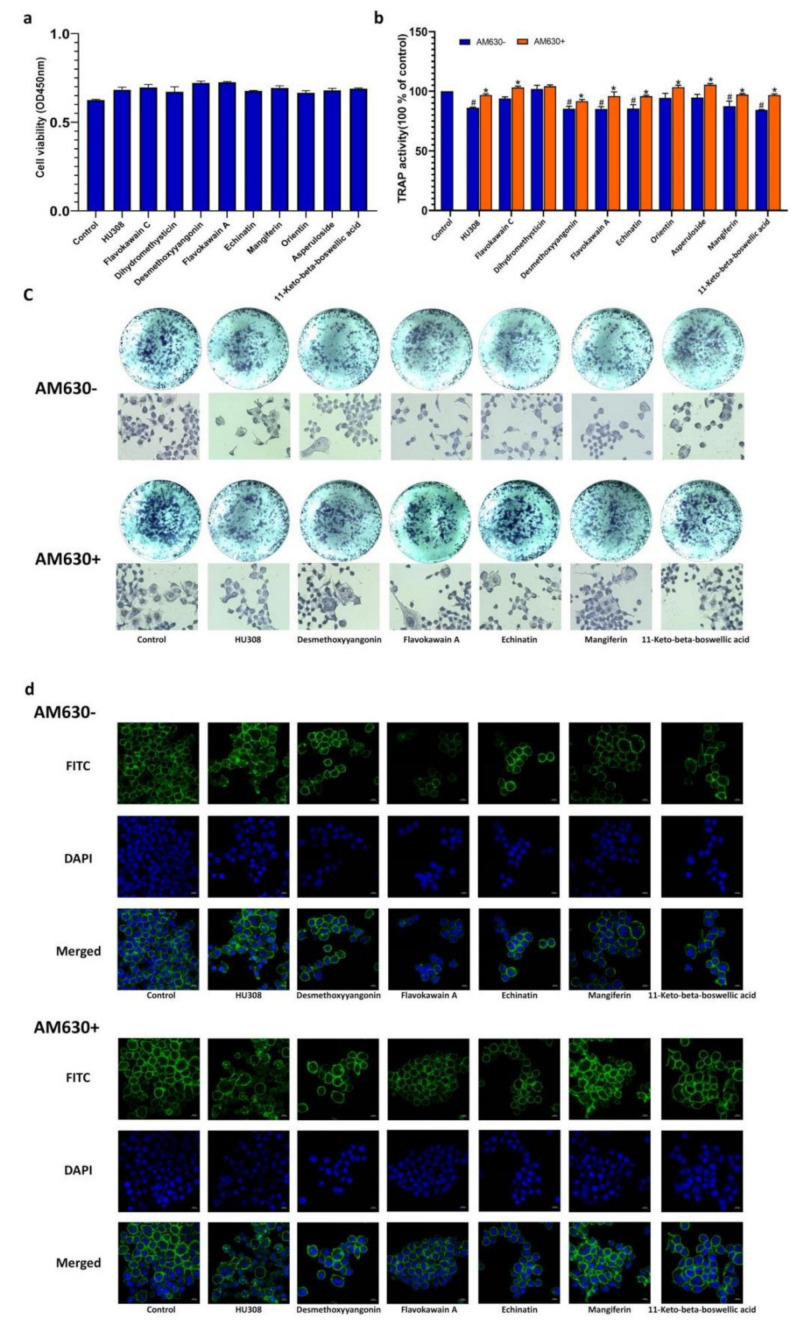
Inhibitory effects of HU308 and screened compounds on the activity of osteoclasts derived from RAW264.7 cells. (**a**) Cell viability. RAW264.7 cells were treated with 1 μM of HU308 or the screened compounds for three days, and the cell viability was detected by CCK8 method. (**b**) The tartrate resistant acid phosphatase (TRAP) activity. After three-day treatment of osteoclasts derived from RAW264.7 cells in the presence or absence of 1 μM AM630, TRAP activity was measured by p-nitrophenyl sodium phosphate method. (**c**) TRAP staining. After three-day treatment of osteoclasts derived from RAW264.7 cells in the presence or absence of 1 μM AM630, TRAP activity was detected with the assay kit. (**d**) F-Actin ring staining. After five-day treatment of osteoclasts derived from RAW264.7 cells in the presence or absence of 1 μM AM630, F-Actin ring was stained with phalloidin and DAPI and then imaged with a fluorescence microscope. The data shown are from three independent experiments and expressed as mean ± SD (*n* = 3). # *p* < 0.05 compared with the control. * *p* < 0.05 compared with corresponding compounds treatment group. TRAP, tartrate resistant acid phosphatase.

**Figure 8 molecules-27-00702-f008:**
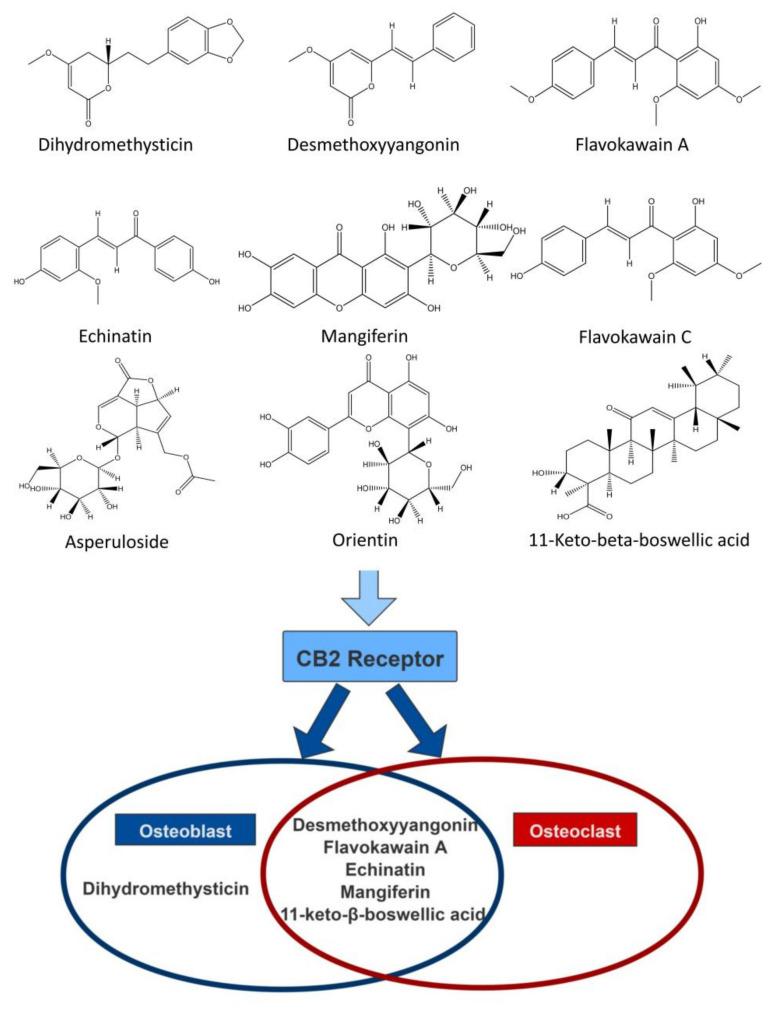
The schematic diagram shows the plant monomers that can further promote osteogenesis (blue circle) and inhibit osteoclast differentiation (red circle) through the CB2 pathway in natural compounds that activate CB2R.

**Table 1 molecules-27-00702-t001:** Interactions and docking scores of HU308 and nine screened compounds with CB2 protein 5ZTY within 4 Å.

S. No	Compound	Docking Score	Amino Acid Residues within 4 Å	No. of Hydrogen Bonds	Hydrogen Bond Interactions	π-π Interactions
1	HU308	−11.604	TYR25, PHE87, SER90, PHE91, PHE94, HIS95, PHE106, ILE110, VAL113, THR114, PHE117, LEU182, PHE183, PRO184, TRP194, TRP258, VAL261, LEU262, MET265, PHE281, ALA282, SER285, CYS288	1	LEU182	PHE87, PHE183
2	Dihydromethysticin-1	−8.053	PHE87, PHE91, HIS95, ILE110, VAL113, THR114, PHE117, PHE183, TRP194, TRP258, VAL261, LEU262, MET265, PHE281, SER285, CYS288	0	-	PHE117
3	Desmethoxyyangonin	−7.985	ILE110, VAL113, THR114, PHE117, PHE183, ILE186, TYR190, LEU191, TRP194, LEU195, TRP258, VAL261, LEU262, MET265, CYS288	0	-	PHE117, PHE183, TRP194
4	Flavokawain A	−8.508	TYR25, ILE27, PHE87, PHE91, PHE94, HIS95, PHE106, ILE110, VAL113, PHE117, LEU182, PHE183, PRO184, TRP194, TRP258, VAL261, MET265, LYS278, PHE281, ALA282, SER285, CYS288	1	SER285	HIS95
5	Echinatin	−8.926	TYR25, ILE27, PHE87, PHE91, HIS95, ILE110, VAL113, THR114, PHE117, LEU182, PHE183, PRO184, TRP194, VAL261, LYS278, PHE281, ALA282, SER285, CYS288	2	THR114, SER285	HIS95, PHE183
6	Mangiferin	−13.968	TYR25, PHE87, SER90, PHE91, PHE94, HIS95, PHE106, ILE110, VAL113, THR114, PHE117, THR118, LEU182, PHE183, PRO184, TRP194, TRP258, VAL261, PHE281, ALA282, SER285, CYS288	3	SER90, LEU182, SER285	PHE87, PHE183
7	11-Keto-beta-boswellic acid	−6.477	TYR25, PHE87, SER90, PHE91, PHE94, HIS95, PHE106, ILE110, VAL113, THR114, PHE117, LEU182, PHE183, PRO184, TRP194, VAL261, LEU262, MET265, PHE281, SER285, CYS288	1	SER285	-
8	Flavokawain C	−7.769	TYR25, ILE27, PHE87, SER90, PHE91, PHE94, HIS95, PHE106, LYS109, ILE110, VAL113, THR114, PHE117, LEU182, PHE183, TRP194, VAL261, MET265, LYS278, PHE281, ALA282, SER285	1	LYS278	PHE94, HIE95
9	Orientin	−14.469	TYR25, ILE27, PHE87, SER90, PHE91, PHE94, HIS95, PHE106, LYS109, ILE110, VAL113, THR114, PHE117, LEU182, PHE183, TRP194, VAL261, MET265, LYS278, PHE281, ALA282, SER285	3	TYR25, PHE95, TPR114	PHE94, PHE183
10	Asperuloside	−12.473	PHE87, SER90, PHE91, PHE94, HIS95, PHE106, ILE110, VAL113, THR114, PHE117, LEU182, PHE183, PRO184, ILE186, TYR190, LEU191, TRP194, VAL261, MET265, PHE281, SER285, CYS288	2	SER90, LEU182	-

**Table 2 molecules-27-00702-t002:** The results of an MD analysis of HU308 and screened compounds.

S. No	Compound	T _stable_ of the Protein (ns)	RMSD Value of the Protein _stable_ (Å)	T _stable_ of the ligand (ns)	RMSD Value of the Ligand _stable_ (Å)
1	HU308	1~10	3.0	4~10	1.7
2	Dihydromethysticin	8~10	3.0	2~10	3.5
3	Desmethoxyyangonin	4~9	3.2	4~10	3.0
4	Flavokawain A	4~10	3.0	4~10	2.6
5	Echinatin	2~10	2.8	1~10	2.6
6	Mangiferin	4~10	3.9	4~10	3.3
7	11-Keto-beta-boswellic acid	6~9	3.2	4~9	2.2
8	Flavokawain C	7~10	3.2	4~8	2.5
9	Orientin	1~10	3.0	1~10	7.0
10	Asperuloside	0.2~8	2.8	6~8.5	3.8

## Data Availability

The data presented in this study are available in the article.

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
