# Peer review of "Identification of Novel Cannabinoid CB2 Receptor Agonists from Botanical Compounds and Preliminary Evaluation of Their Anti-Osteoporotic Effects"

_molecules, 2022, doi:10.3390/molecules27030702_

Round 1
Reviewer 1 Report
The manuscript by Hu and co-workers describe the identification and evaluation of 10 natural products as novel CB2 agonists. This is an interesting work, the results are important for the CB2-research field and I recommend the publication of the manuscript after the following points are addressed.
The English has to be improved. Some sentences are too long and hard to read.
Title: I would replace “analysis” with “preliminary evaluation”
Abstract:
e.g. second sentence from the abstract should be either divided in 2-3 sentences or it should be modified as: “ In present study we 1) established… 2) validate… 3)…”
“Ten compounds were screened from 90 botanical compounds as CB2 agonist,” this means that you had 90 botanical compounds and only 10 were screened, if this is not the case then pls rephrase.
Are you sure that “Good affinity as shown by cAMP assay” is correct? cAMP assay is a functional assay and not a binding assay (displacement), as a result it is not affinity but potency what you are obtaining, from molecular docking is indeed something similar to affinity.
Manuscript:
Line 39: delete “And” from the start of the sentence
Line 55: “and even incidence of osteoporosis” sounds awkward, rephrase
Line 72-73 “Their wide availability, easy access, high lipophilicity and less side effects make them excellent candidates for therapeutic intervention” is not quite correct, e.g. : Easy access: many natural compounds occur in very low amounts a to isolate in pure form, large amounts of plant is needed and sophisticated purification steps, on the other hand, chemical synthesis can deliver the compounds in high amounts and sometimes quite accessible chemical routes.
“less side effects” this is available only for a number of natural products, many other natural products are produced by the plant as part of the defense system against predators and thus toxic (e.g. THC, nicotin).
Line 74-78, rephrase, it sounds like you screened 10 compounds but you had 90 in your stock.
In the introduction the screening of 90 compounds is mentioned, on line 132 and 341 only 75 are mentioned, the rest were not screened? Then why to mention them?
Line 329, check literature cited Zhang [26]
Conclusion:
Line 641-642: molecular docking is not demonstrating but predicting or estimating.
Author Response
Thank you for your letter and for the reviewer’s useful comments and suggestions concerning our manuscript entitled “Identification of novel cannabinoid CB2 receptor agonists from botanical compounds and preliminary evaluation of their antiosteoporotic effects” (molecules-1536408). Those comments are all valuable and helpful for revising and improving our paper, as well as the important guiding significance to our researches. We have modified the manuscript accordingly, revised portion are marked in red in the manuscript. The detailed corrections are listed below point by point:
Point 1: The English has to be improved. Some sentences are too long and hard to read.
Response 1: Thanks for reviewer’s suggestion, and we have checked and improved the English expression in the manuscript, and also polished the language by a native English speaker.
Point 2: I would replace “analysis” with “preliminary evaluation”.
Response 2: Thanks for reviewer’s suggestion, and we have replaced “analysis” with “preliminary evaluation”.
Point 3: e.g. second sentence from the abstract should be either divided in 2-3 sentences or it should be modified as: “In present study we 1) established; 2) validate; 3)…”.
Response 3: As reviewer’s suggested that we have modified the above sentences.
Point 4: Ten compounds were screened from 90 botanical compounds as CB2 agonist, this means that you had 90 botanical compounds and only 10 were screened, if this is not the case then pls rephrase.
Response 4: Concerning this issue, we need to give an explanation. Actually, we screened CB2 agonists from 90 botanical samples, including 75 compounds and 15 extracts. In the revising the manuscript, we found that 6 botanical compounds isolated in our laboratory have not been published owing to their chemical configuration not being elucidate. Therefore, we deleted the results of these 6 compounds in the present manuscript, and reported the results of 69 compounds. So, the above sentences has been revised as ”Nine CB2R agonists were identified by screening from 69 botanical compounds”.
Point 5: Are you sure that “Good affinity as shown by cAMP assay” is correct? cAMP assay is a functional assay and not a binding assay (displacement), as a result it is not affinity but potency what you are obtaining, from molecular docking is indeed something similar to affinity.
Response 5: As reviewer’s pointed out that we have revised the above sentences as “these CB2R agonists exhibited remarkable inhibitory effects on cAMP accumulation, and good affinity to CB2R as evidenced by molecular docking and molecular dynamics”.
Point 6: Line 39: delete “And” from the start of the sentence
Response 6: The “And” have been deleted from that sentence.
Point 7: Line 55: “and even incidence of osteoporosis” sounds awkward, rephrase
Response 7: We have revised the “and even incidence of osteoporosis” as “even the osteoporotic fracture”.
Point 8: Line 72-73 “Their wide availability, easy access, high lipophilicity and less side effects make them excellent candidates for therapeutic intervention” is not quite correct, e.g. : Easy access: many natural compounds occur in very low amounts a to isolate in pure form, large amounts of plant is needed and sophisticated purification steps, on the other hand, chemical synthesis can deliver the compounds in high amounts and sometimes quite accessible chemical routes.
Point 9: “less side effects” this is available only for a number of natural products, many other natural products are produced by the plant as part of the defense system against predators and thus toxic (e.g. THC, nicotin).
Response 8, 9: Point 8 and 9 are about the same sentence, and we revised the above sentence as “Some botanical compounds have been developed into clinical medicines such as artemisinin, paclitaxel, vincristine and morphine”.
Point 10: Line 74-78, rephrase, it sounds like you screened 10 compounds but you had 90 in your stock.
Response 10: Thanks for reviewer’s comments, and the above sentence have been revised as “we developed a double luciferase screening system to screen CB2R agonists from 69 botanical compounds,”.
Point 11: In the introduction the screening of 90 compounds is mentioned, on line 132 and 341 only 75 are mentioned, the rest were not screened? Then why to mention them?
Response 11: This query in identical with “point 3”, and we explained and revised as follows: Actually, we screened CB2 agonists from 90 botanical samples, including 75 compounds and 15 extracts. In the revising the manuscript, we found that 6 botanical compounds isolated in our laboratory have not been published owing to their chemical configuration not being elucidated. Therefore, we deleted the results of these 6 compounds in the present manuscript, and reported the results of 69 compounds. So, the above sentences has been revised as “Nine CB2R agonists were identified by screening from 69 botanical compounds”.
Point 12: Line 329, check literature cited Zhang [26]
Response 12: We have corrected and revised the above reference.
Point 13: Conclusion: Line 641-642: molecular docking is not demonstrating but predicting or estimating.
Response 13: Thanks, and we have revised “demonstrated” as “predicted”.
Reviewer 2 Report
CB2 receptors are getting more and more attention in the central nervous and peripheral system. The authors established a screening system and identified 10 CB2 agonists from 90 botanical compounds, validated their agnostic activities on CB2 using cAMP levels, molecular docking, and molecular dynamics simulations, and then further evaluated their antiosteoporotic activities on osteoblast and osteoclast. This may provide novel leads for the discovery of antiosteoporotic drugs. In addition, identifying novel cannabinoid ligands also can provide novel mechanisms of action for the known therapeutic effects of these compounds.
The experimental design is rigorous, and the results are reliable, which are sufficient to support the author's point. However, there are errors in language writing and spelling, including whether there is a space between the value and the unit to be unified, and whether it is bold. Further carefully checking and optimizing the writing is required.
Author Response
Thank you for your letter and for the reviewer’s useful comments and suggestions concerning our manuscript entitled “Identification of novel cannabinoid CB2 receptor agonists from botanical compounds and preliminary evaluation of their antiosteoporotic effects” (molecules-1536408). Those comments are all valuable and helpful for revising and improving our paper, as well as the important guiding significance to our researches. We have modified the manuscript accordingly, revised portion are marked in red in the manuscript. The detailed corrections are listed below point by point:
Point 1: The experimental design is rigorous, and the results are reliable, which are sufficient to support the author's point. However, there are errors in language writing and spelling, including whether there is a space between the value and the unit to be unified, and whether it is bold. Further carefully checking and optimizing the writing is required.
Response 1: Thanks for reviewer’s comments, and we have checked and revised the English language writing and spelling throughout the manuscript, and polished the the English language by a native speaker.
Reviewer 3 Report
The manuscript by Si-jing Hu et al., describes a double luciferase screening system for CB2 receptor modulators, and it reports the agonistic activities of 20 natural compounds on CB2 receptor by determining cAMP accumulation using HEK293 cells. Moreover, the regulatory effects of the most active compounds on bone metabolism in osteoblast and osteoclast were evaluated. This manuscript could be of interest in the field. However, I have some comments that might increase the impact of this manuscript and help the reader navigate this complex field
- pg 2 line 95: explain the formula, Z’=1−(3σs+3σc)/(μs−μc). Please Introduce an appropriate reference.
-pg 3 line 108: Authors need to be careful of their terminology. AM630 is CB2 receptor inverse agonists. A pure antagonist for CB2R has yet to be developed.
-pg3 line 107:…. “HU-308 could significantly increase the relative luciferase activity”. Why did you use “could”?
-pg 3 line 108: agnostic? Check carefully the manuscript to eliminate typing errors.
-Please give the full name of EGFP when it is used for the first time.
-Please indicate the purity grade of each compound tested.
-Are you sure that the structure reported in figure 8 is humuluslupulus E? I didn’t find it in literature.
-pg 4 line 138. It is not correct to use the term “specificity” because these compounds could be ligands also for other targets. In particular, are you sure that these compounds are not ligands for CB1 receptors? Several natural cannabinoid have very mixed pharmacology.
- Please change CB2 into CB2 receptor
- the activity of HU308 on osteoblasts and osteoclasts that is reported in the literature, as well as the activity of some natural cannabinoids should be discussed.
Author Response
Thank you for your letter and for the reviewer’s useful comments and suggestions concerning our manuscript entitled “Identification of novel cannabinoid CB2 receptor agonists from botanical compounds and preliminary evaluation of their antiosteoporotic effects” (molecules-1536408). Those comments are all valuable and helpful for revising and improving our paper, as well as the important guiding significance to our researches. We have modified the manuscript accordingly, revised portion are marked in red in the manuscript. The detailed corrections are listed below point by point:
Point 1: -pg 2 line 95: explain the formula, Z’=1−(3σs+3σc)/(μs−μc). Please Introduce an appropriate reference.
Response 1: We have explain the formula of Z’ factor calculation, and adjusted its detailed description into the section of 4.3.5, and also cited a reference in the manuscript.
Point 2: -pg 3 line 108: Authors need to be careful of their terminology. AM630 is CB2 receptor inverse agonists. A pure antagonist for CB2R has yet to be developed.
Response 2: As reviewer suggested that we have revised “antagonist” as “inverse agonist” throughout the manuscript.
Point 3: -pg3 line 107: “HU-308 could significantly increase the relative luciferase activity”. Why did you use “could”?
Response 3: We are agree to reviewer’s suggestion, and deleted “could” in the above sentence.
Point 4: -pg 3 line 108: agnostic? Check carefully the manuscript to eliminate typing errors.
Response 4: Thanks for reviewer’s comments, and we have checked and revised the English language writing and spelling throughout the manuscript, and polished the the English language by a native speaker.
Point 5: -Please give the full name of EGFP when it is used for the first time.
Response 5: We have supplemented the full name of EGFP in the manuscript.
Point 6: -Please indicate the purity grade of each compound tested.
Response 6: The purity grade of tested compounds are more than 98% as analyzed by HPLC method, and these information have been added in the manuscript.
Point 7: -Are you sure that the structure reported in figure 8 is humuluslupulus E? I didn’t find it in literature.
Response 7: I am sorry, this is our carelessness. The compound humuluslupulus E is isolated from Humulus lupulus L. in our laboratory. But we have not elucidated the chemical configuration of this compound, so the paper was not published until now. Therefore, we deleted the results of humuluslupulus E in this manuscript.
Point 8: -pg 4 line 138. It is not correct to use the term “specificity” because these compounds could be ligands also for other targets. In particular, are you sure that these compounds are not ligands for CB1 receptors? Several natural cannabinoid have very mixed pharmacology.
Response 8: We have replaced “specificity” with “agonistic effects”, and the whole sentence has been changed into “Then, the agonistic effect of these compounds on CB2R was further verified by using AM630, a CB2R inverse agonist.”.
Point 9: -Please change CB2 into CB2 receptor
Response 9: Thanks for your suggestion, and we have revised “CB2” as “CB2 receptor” or its abbreviation “CB2R”.
Point 10: -the activity of HU308 on osteoblasts and osteoclasts that is reported in the literature, as well as the activity of some natural cannabinoids should be discussed.
Response 10: Thanks for your suggestion, and we had discussed the effects of HU308 and other natural cannabinoids on osteoblasts and osteoclasts in the section of discussion. It is as follows: HU308 as a selective CB2R agonist has been shown to decrease the bone loss in osteoporotic animal model, such as ovariectomized rats, and spinal cord injury rats. In addition, it also exerted a regulatory effect on osteoblasts and osteoclasts in a CB2R-dependent manner. Furthermore, some CB2R agonist from natural products, such as (E)-β-caryophyllene, magnolol, 4-O-methylhonokiol, celastrol, resveratrol, betulinic acid, anthocyanins and biochanin A have been shown to have potential effects on bone metabolism. These findings , together with our results, provide the potential lead for for anti-osteoporotic drugs targeting CB2R.
Round 2
Reviewer 3 Report
The suggested changes have been introduced by the authors. For that, I think this work can be published.